# The Perspective of the General Population in Saudi Arabia towards Do-Not-Resuscitate (DNR) Orders: A Cross-Sectional Study

**DOI:** 10.3390/healthcare11142073

**Published:** 2023-07-20

**Authors:** Rayan Abubakker Qutob, Najd Khalid Aljarba, Bassam Abdulaziz Alhusaini, Omar Nasser Alzaid, Abdullah Hussien Alghamdi, Abdullah Abdulaziz Alaryni, Abdullah Ibraheem Bukhari, Ahmed Alburakan, Abdulrahman Mohammed Alanazi, Eysa Nahar Alsolamy, Omar Abdulaziz Alfozan, Saad Abdullah Alzmamy, Abdalmohsen A Ababtain, Alyaa Elhazmi, Osamah A. Hakami

**Affiliations:** 1College of Medicine, Imam Mohammad Ibn Saud Islamic University, Riyadh 11432, Saudi Arabia; dr.rayanq@hotmail.com (R.A.Q.); bam.h98s@gmail.com (B.A.A.); alzaidomar23@gmail.com (O.N.A.); dr.alhomrani@gmail.com (A.H.A.); al3raini@hotmail.com (A.A.A.); bukhariai@hotmail.com (A.I.B.); amn654@gmail.com (A.M.A.); eysa783@hotmail.com (E.N.A.); omaralfozan00000@gmail.com (O.A.A.); alzmamysaad@gmail.com (S.A.A.); ohakami.md@gmail.com (O.A.H.); 2Department of Surgery, College of Medicine, King Saud University, Riyadh 11451, Saudi Arabia; aalburakan@ksu.edu.sa; 3Critical Care Department, King Abdullah bin Abdulaziz University Hospital, Princess Nourah bint Abdulrahman University, Riyadh 11451, Saudi Arabia; aaababtain@kaauh.edu.sa; 4College of Medicine, AlFaisal University, Riyadh 11533, Saudi Arabia; a.m.haz@live.com

**Keywords:** attitude, knowledge, do-not-resuscitate, DNR, Saudi Arabia

## Abstract

Objectives: In the event of cardiac arrest, cardiopulmonary resuscitation (CPR) is an emergency procedure used to maintain the heart and lungs functional simultaneously. The do-not-resuscitate (DNR) order prohibits CPR and is therefore legally required. Despite this, a DNR remains a delicate and contentious issue that places physicians in morally ambiguous situations. This study aimed to assess Saudi citizens’ understanding of DNR orders, prior exposure to them, and preferences for DNR conversations. Methods: This was an online cross-sectional study that was conducted between January and April 2023 and aimed to assess the knowledge regarding DNR orders among Saudi populations. This study adapted a previously developed questionnaire tool by Al Ahmadi et al., which examined the knowledge and attitude toward do-not-resuscitate among patients and their relatives visiting outpatient clinics. Binary logistic regression analysis was the mean knowledge score for the study participants. Results: A total of 920 participants were involved in this study. Almost half of the study participants (49.6%) reported that they had heard of DNR before. The most commonly reported source of their information on DNR was healthcare providers (58.2%). The mean knowledge score of the study participants was 1.9 (1.3) out of 6, which is equal to 31.7% of the total maximum score. This demonstrates the weak level of knowledge about DNR among the general public. Females, divorced, and those who had a post-graduate level of education were more likely to be knowledgeable of DNR compared to others (*p* < 0.05). Around one-quarter of the study participants showed agreement with DNR. More than half of them (59.5%) believe that physicians should be involved in DNR decision making. Conclusions: Saudi Arabia’s general community has limited knowledge of DNR. It is recommended that healthcare professionals increase patients’ and caregivers’ understanding of this concept. This will improve the planning and the provision of end-of-life care.

## 1. Introduction

Cardiopulmonary resuscitation (CPR) is a life-saving emergency procedure used to keep the heart and lungs functioning at the same time in the case of cardiac arrest [1]. The likelihood of survival can be increased by a factor of two or three by performing CPR right away [2]. Kouwenhoven has accepted the idea of CPR since 1960. The method was first referred to as a closed-chest cardiac massage [3]. All cardiac arrest patients are required to receive emergency care as of 1980 [4]. However, as the patient’s prognosis has a substantial impact on CPR’s effectiveness, it is thought that this approach should only be suggested for patients with a good chance of survival [5]. Studies showed that only 10–15 percent of patients were alive at discharge after receiving CPR. Thirty years have passed with no change in these numbers [5]. The do-not-resuscitate (DNR) order forbids cardiopulmonary resuscitation and is therefore legally obligatory. Despite this, a DNR continues to be a delicate and contentious topic that puts doctors in morally confusing circumstances [6], particularly in Islamic countries [7]. Globally, legal, social, cultural, and religious aspects have a big effect on DNR [8]. In Saudi Arabia, the role of religion in determining policy is prominent. In 1988, the Director General of Scientific Research and Iftah, an approved source of legislation, established the first national DNR regulations. Based on Fatwa number 12,086, they claimed that the professional decision to DNR was acceptable, regardless of the preferences of the patients or their families. Furthermore, it says that “there is no need to use resuscitators or life-supporting machines if three knowledgeable and trustworthy physicians agree that it is not appropriate to resuscitate a patient when it is evident that they are suffering from an intractable, incurable illness and that death is inevitable”. This strategy has been adopted by hospitals, but it has not yet been put into practice [7].

Few studies have been undertaken in the majority of Arab and Muslim countries, despite the fact that knowledge and attitudes towards DNR have been thoroughly researched in Western populations [9]. In Saudi Arabia, there have been few studies that examined perception towards DNR, with the majority among patients with specific diseases such as cancer or exploring the perceptions of healthcare professionals [10,11,12,13]. A cross-sectional study was carried out in Saudi Arabia in 2017 to look at how the general public and the loved ones of cancer patients felt about the DNR order. They found that having a background in medicine was significantly connected with accepting a DNR order, and that 58.6% of people who knew about the DNR practice were employed in the medical industry. In addition, 76% of the individuals had false assumptions about DNR usage. In terms of decision-making, 69.9% of participants said they wanted a say in whether they were revived or not. They came to the conclusion that most participants were unaware of the DNR practice [13]. Another study in Saudi Arabia conducted at King Fahad Medical City among outpatient visitors revealed that the majority of participants (75%) were familiar with DNR, but only half could correctly define it [14].

Therefore, this study aimed to assess Saudi citizens’ understanding of DNR orders, prior exposure to them, and preferences for DNR conversations. Examining the public’s perception of DNR is crucial because it influences legislation and guidelines, informs end-of-life decision-making, and improves communication between healthcare providers and patients.

## 2. Methods

### 2.1. Study Design and Settings

This was an online cross-sectional study that was conducted between January and April 2023 and aimed to assess knowledge regarding DNR orders among Saudi populations. This study employed the non-probability convenience sampling technique. Data collection was carried out by the co-authors using an online-based questionnaire that was distributed through social media websites (Facebook, Instagram, and Snap Chat). The link was posted on the homepage of the data collectors, asking participants who meet the inclusion criteria to participate in the study. Furthermore, they were asked to send the study link to their friends and colleagues.

### 2.2. Sample Population

The general population in Saudi Arabia from the age of 18 years old and over.

### 2.3. Questionnaire Tool

This study adapted a previously developed questionnaire tool by Al Ahmadi et al., which examined the knowledge and attitude toward do-not-resuscitate among patients and their relatives visiting outpatient clinics [15]. Our questionnaire asked the participants about their demographic characteristics. This was followed by exploring their attitude and knowledge of DNR. Six questions were used to estimate the knowledge scores of the study participants. For each correct answer, the participants were given a score of one. The maximum score was six. The higher the score, the better the knowledge of DNR. We defined any score below the midpoint of the maximum obtainable score (which was six) as poor knowledge.

### 2.4. Questionnaire Translation

Questionnaire translation to the Arabic language was based on the forward–backward technique, in which the translator independently focused on preserving the actual meaning of the questionnaire items rather than on word-by-word translation.

### 2.5. Face Validity Check and Piloting Phase

The Arabic-translated version of the questionnaire was checked by expert clinicians for clarity and comprehensibility. They confirmed that the questionnaire items are clear and support the study objectives. This was followed by conducting a small pilot study on a group of participants from the general public. The questionnaire pilot study verified the following: the clarity and understandability of the survey items, their relevance and appropriateness to the research objectives, the appropriateness of the length of the questionnaire and the time required for completion, and the appropriateness of the response options and scale used.

### 2.6. Statistical Analysis

Continuous data were presented as mean and standard deviation (SD) as the data were normally distributed and checked using the histogram, skewness, and kurtosis measures. Categorical variables were presented using frequency and percentage. A binary logistic regression analysis was used to identify predictors of better knowledge of DNR. The cut-off point to identify the dummy variable for the binary logistic regression analysis was the mean knowledge score for the study participants. A 95% confidence interval (*p* ≤ 0.05) was applied to indicate the statistical significance of the results, and a significance level of 5% was assigned. All data were analyzed using the Statistical Package for Social Science software (version 27).

## 3. Results

### 3.1. Participants’ Baseline Characteristics

Table 1 presents the baseline characteristics of the study participants. A total of 920 participants were involved in this study. Around 44.0% of the participants were aged 21–30 years. More than half of them (65.0%) were males and reported that they held a bachelor’s degree (62.2%). Around 29.0% of them were living in the Western region of Saudi Arabia. Almost half of them (49.7%) were married. Almost half of them (47.1%) reported that they are employed.

### 3.2. Knowledge of and Attitude towards Do-Not-Resuscitate

Table 2 presents the responses of study participants concerning their knowledge of DNR questions. Almost half of the study participants (49.6%) reported that they had heard of DNR before. The most commonly reported source of their information on DNR was healthcare providers (58.2%). When the participants were asked about the correct definition of DNR, around 44.0% defined it as “It is an order made by the physician and reported to all healthcare providers that if a patient’s heart or breathing stops, not to perform CPR because it is futile, regardless of the age of the patient”.

Almost one-fifth of the study participants (18.9%) reported that they had had previous experience with DNR, of which 73.3% were with first-degree relatives. Around one-quarter of the study participants (25.0%) reported that there is a clear policy regarding DNR in Saudi Arabia, and 18.0% reported that there is a fatwa (a formal ruling or interpretation on a point of Islamic law given by a qualified legal scholar) regarding DNR in Saudi Arabia.

### 3.3. Predictors of Better Knowledge about DNR

Table 3 below presents predictors of better knowledge of DNR. The mean knowledge score of the study participants was 1.9 (SD: 1.3) out of 6, which is equal to 31.7% of the total maximum score. This demonstrates the weak level of knowledge about DNR among the general public. Females, divorced, and those who had a post-graduate level of education were more likely to be knowledgeable of DNR compared to others (*p* < 0.05).

### 3.4. Attitude towards Do-Not-Resuscitate

Table 4 presents participants’ responses to questions that examined their attitude towards DNR. Around one-quarter of the study participants showed agreement with DNR. More than half of them (59.5%) believed that physicians should be involved in DNR decision making. The vast majority of the participants (88.3%) reported that parents have the right to know about DNR decisions. More than half of the study participants (58.7%) reported that the patient has the right to reject the DNR decision. The vast majority of the participants (88.2%) reported that religious concerns are the most important factors to be considered when making the DNR decision.

## 4. Discussion

The key findings of this study are: (1) almost half of the study participants reported that they had heard of DNR before; (2) a weak level of knowledge about DNR among the general public was identified in this study; (3) females, divorced people, and those who have a post-graduate level of education were more likely to be knowledgeable about DNR compared to others; and (4) around one-quarter of the study participants showed agreement with DNR.

In our study, nearly half of the participants (49.6%) reported that they were familiar with DNR. When asked for the correct definition of DNR, approximately 44.0% of the participants stated, “It is an order issued by a physician and communicated to all healthcare providers stating that if a patient’s heart or breathing stops, CPR should not be performed because it is futile, regardless of the patient’s age.” This was lower than the findings of a previous study that was conducted at King Fahad Medical City among visitors to the outpatient settings, which revealed that 75% of participants were familiar with DNR but only 50% could correctly define it [14].

In our study, the mean knowledge score of the study participants was 1.9 (1.3) out of 6, or 31.7% of the highest score possible. This demonstrates how little the broader public knows about DNR. This was much lower than the findings of a previous study that was conducted by Kaneetah et al. in Saudi Arabia, where the average score of the study participants was equal to 71.4% [13]. A possible justification for the differences between studies could be the difference in demographic characteristics between study participants, as Kaneetah et al.’s study examined the perspectives of the general public and relatives of cancer patients [13]. Lack of education and awareness is one of the primary causes of the public’s lack of DNR knowledge. In addition, limited healthcare provider–patient communication due to communication difficulties, time constraints, or cultural barriers may be significant contributors to the general public’s lack of DNR knowledge [16,17,18].

In our study, females, divorced individuals, and those with a postgraduate degree were more likely than others to be familiar with DNR (*p* < 0.05). In a previous study, Al Ahmadi et al. found that participants familiar with DNR had a high level of education, but there were no significant differences in gender, age, or marital status [15]. Another study by Vilpert et al. reported that females are better informed about end-of-life care options than males [19]. Additionally, Vilpert et al. found that individuals with higher education levels have better knowledge about end-of-life care options [19]. Multiple factors could influence the public’s knowledge about DNR, such as gender, knowledge, and marital status. Higher levels of education (such as postgraduate degrees) are frequently accompanied by greater exposure to healthcare-related information. As a result, access to education and information, as well as health literacy, improves [19]. Other contributing factors may include personal experience, life circumstances, and social factors, and the fact that females, divorced individuals, and those with higher levels of education could have greater opportunities to discuss DNR.

In our study, 58.2% of the study participants reported that healthcare professionals were their main source of information about DNR. A previous cross-sectional study was carried out in Jeddah, Saudi Arabia, in 2018 with the purpose of evaluating the knowledge and attitudes of patients and their families who visited outpatient clinics at King Abdulaziz University Hospital about DNR. Around one-quarter (26.3%) of the study participants were familiar with the phrase DNR, 44.8% chose the right meaning, 5.2% had used it before, and 34.3% reported that social media was the main source for their information about DNR. The majority of respondents were unaware of a DNR policy or fatwa (a legal interpretation on an issue of Islamic law). They came to the conclusion that there is a serious information gap regarding DNR in our culture [15]. The primary source of information on DNR should be healthcare providers. Insufficient training and a lack of time, among other factors, were determined to be the most frequent impediments to addressing DNR with patients or their families by several researchers who have investigated these issues [9,18,20].

A previous cross-sectional study was carried out in Canada in 2009 to assess outpatients’ knowledge and comprehension of DNR orders. They found that 84% of people were familiar with the phrase DNR. One of the study’s most important findings was that 86% of respondents preferred to talk about DNR choices with family doctors over other people. Additionally, 56% of respondents said that the initial DNR conversations should be held when the subject is still in good health [21].

Confirming previous study findings, our participants exhibited a favorable attitude toward organ donation and a conservative approach with DNR patients, which reflects the Islamic and cultural values ingrained in their culture [15]. A previous systematic review of Saudis’ knowledge, attitudes, and perspectives regarding DNR orders revealed that religious issues and the risk of vegetative conditions were the most influential factors in DNR decisions [16]. The study by Abdullah Amoudi and his group set out to evaluate the knowledge and attitudes of the interns and residents about DNR and its implications for patient care. The study found that there is a lack of understanding about DNR laws, the fatwa, and how to care for patients who have been declared DNR. Furthermore, The majority (66%) of internal medicine doctors had DNR conversations with patients, family members, or surrogate decision makers. Faculty members, who observed 43% of these talks, reported feeling at ease in them 51.9% of the time. The majority of the population thought that taking part in more educational programs would improve their capacity to handle DNR-related concerns [11].

The opinions of both medical and non-medical students about the DNR choice and the underlying variables were also evaluated in a study carried out at a Hong Kong university in 2005. They found that 74% of participants thought the patient’s desires were the most important thing the healthcare staff should take into account [22]. Muneerah Albugami et al. (2017) discussed how medical residents use DNR orders and how they see them [10]. The vast majority of the study participants (85.6%) reported that doctors should not recommend diagnostic treatments for DNR patients, nor should they give them blood or antibiotics. Furthermore, 36.8% of respondents said that the practitioner alone should decide whether to use DNR or not. Around 43.9% of respondents thought the patient and doctor should determine whether to issue a DNR order. Around 87.7% of respondents agreed that discussing DNR during formal training is necessary [10]. In addition, the vast majority of residents had misunderstandings about comfortable care and DNR patient care in general [10]. Another study that examined medical students’ and interns’ knowledge of and attitudes toward DNR found that more than half of the participants (58.3%) had not attended any DNR-related lectures or sessions, and the majority of participants (73.2%) were aware of the DNR directive. A sizable number of medical students thought that participating in a lecture or discussion about DNR would improve their capacity to speak with patients and their families about the subject. More than half of the respondents (55%) thought that the DNR was governed by an Islamic Fatwa [23]. Another study aimed to find out nurses’ and doctors’ viewpoints on the DNR decision-making process. According to the study’s findings, a medical recommendation has to be codified, and the decision-making and implementation procedures need to be made clearer. The policy must also outline the obligations of the doctor, nurse, patient, and other members of the medical team [12]. Around one-quarter of the study participants showed agreement with DNR. More than half of them (59.5%) believed that physicians should be involved in DNR decision making. The vast majority of the participants (88.3%) reported that parents have the right to know about DNR decisions [12].

All healthcare professionals are required to discuss DNR and advanced directive status with patients. The objective is to educate the patient and family that DNR does not imply a poor quality of life; in fact, the opposite is true [24]. It is essential to assure the patient’s family that he or she will be made comfortable and that any discomfort will be addressed. In addition to increasing public awareness of DNR, it is crucial to educate medical students and healthcare professionals about DNR for the following reasons: education regarding DNR enables medical students and healthcare professionals to comprehend the ethical principles underlying end-of-life care and make informed and ethical decisions when implementing DNR orders. In addition, increased awareness of DNR improves communication between medical professionals, patients, and their families [20]. In addition, DNR education enhances overall end-of-life care.

This study has some limitations. The cross-sectional study design limited our ability to investigate causality among study variables. The use of the convenience sampling technique and the online survey study design may have impacted the generalizability of our study’s findings, as a substantial proportion of our target population may lack access to social media websites. Nevertheless, according to the most recent available data from 2023, approximately 79.3 percent of the Saudi Arabian population is active on social media. We examined the external validity of the study instrument using a pilot study without performing test–retest reliability or inter-rater reliability tests, which might have an influence on the reliability of the instrument.

## 5. Conclusions

The knowledge of DNR is weak among the general public in Saudi Arabia. Decision makers in the healthcare sector are advised to enhance the public’s knowledge about DNR through multiple channels, such as enhancing medical education by their healthcare professionals and through social media. This will enhance healthcare planning and the provision of end-of-life care.

## Figures and Tables

**Table 1 healthcare-11-02073-t001:** Participants’ baseline characteristics (n = 920).

Variable	Frequency	Percentage
Age categories		
18–20 years	108	11.7%
21–30 years	401	43.6%
31–40 years	142	15.4%
41–50 years	142	15.4%
51–60 years	92	10.0%
61–70 years	35	3.8%
Gender		
Males	596	64.8%
Nationality		
Saudi	846	92.0%
Area of residency		
Southern region	120	13.0%
Northern region	133	14.5%
Central region	251	27.3%
Eastern region	153	16.6%
Western region	263	28.6%
Social status		
Single	440	47.8%
Married	457	49.7%
Divorced	13	1.4%
Widowed	10	1.1%
Education level		
High school level or lower	229	24.9%
Bachelor degree	572	62.2%
Post-graduate level	119	12.9%
Occupation		
Employed	433	47.1%
Unemployed	114	12.4%
Retired	73	7.9%
Students	300	32.6%

**Table 2 healthcare-11-02073-t002:** Knowledge about do-not-resuscitate (n = 920).

Variable	Frequency	Percentage
Have you ever heard about the DNR term? (Yes)	456	49.6%
From where did you hear about DNR? (n = 456)		
Healthcare providers	265	58.2%
Social media	90	19.7%
Relatives and friends	43	9.4%
Internet	58	12.7%
What do you think is the correct definition of do-not-resuscitate (DNR)? (n = 456)		
“It is an order made by the physician and reported to all healthcare providers that if a patient’s heart or breathing stops not to perform CPR because it is futile, regardless the age of the patient”.	199	43.6%
“It is an order that can be requested by the patient, not to perform CPR if the patient’s heart or breathing stops”.	84	18.4%
“It is an order made by the physician and reported to all healthcare providers that if a patient’s heart or breathing stops not to perform CPR because it is futile, with respect to the patient’s decision whether to accept or reject the DNR order”.	74	16.2%
“It is an order requested by physicians for patients with end-stage diseases either by stopping giving medications or withdrawal of life-supporting machines”.	68	14.9%
“It is an order made by the physician and reported to all healthcare providers that if a patient’s heart or breathing stops not to perform CPR because it is futile, for patients aged 50 years or above”.	31	6.8%
Did you have previous experience with DNR? (n= 456) (Yes)	86	18.9%
If yes, with whom? (n = 86)		
First-degree relatives	63	73.3%
Friends	18	20.9%
Spouse	5	5.8%
Do you know if there is a clear policy regarding DNR in Saudi Arabia?		
Yes	230	25.0%
No	110	12.0%
I do not know	580	63.0%
Do you know if there is a fatwa regarding DNR in Saudi Arabia?		
Yes	165	17.9%
No	39	4.2%
I do not know	716	77.8%

**Table 3 healthcare-11-02073-t003:** Predictors of better knowledge about DNR.

Variable	Odds Ratio (95% Confidence Interval)	*p*-Value
Age categories		
18–20 years (reference group)	1.00	
21–30 years	1.26 (0.97–1.64)	0.088
31–40 years	1.06 (0.74–1.53)	0.749
41–50 years	1.03 (0.71–1.47)	0.892
51–60 years	0.89 (0.58–1.38)	0.609
61–70 years	-	-
Gender		
Female (reference group)	1.00	
Males	0.72 (0.55–0.95)	0.020
Nationality		
Non-Saudi (reference group)	1.00	
Saudi	0.84 (0.52–1.36)	0.481
Area of residency		
Southern region (reference group)	1.00	
Northern region	1.20 (0.82–1.75)	0.348
Central region	0.85 (0.63–1.14)	0.266
Eastern region	0.74 (0.52–1.04)	0.085
Western region	1.08 (0.81–1.45)	0.591
Social status		
Single (reference group)	1.00	
Married	1.06 (0.82–1.38)	0.665
Divorced	8.89 (1.15–68.66)	0.036
Widowed	0.72 (0.21–2.52)	0.611
Education level		
High school level or lower (reference group)	1.00	
Bachelor degree	0.99 (0.76–1.30)	0.958
Post-graduate level	1.65 (1.09–2.48)	0.017
Occupation		
Employed (reference group)	1.00	
Unemployed	0.64 (0.43–0.95)	0.026
Retired	0.77 (0.48–1.25)	0.290
Students	1.03 (0.78–1.36)	0.865

**Table 4 healthcare-11-02073-t004:** Attitude towards do-not-resuscitate.

Variable		Frequency	Percentage
Degree of agreement with DNR:			
Agree		230	25.0%
Neutral		314	34.1%
Disagree		376	40.9%
Who do you think should be involved in DNR decision making?			
Physician		547	59.5%
First-degree relatives		222	24.1%
Patient		151	16.4%
Who do you think have the right to know about DNR decision (you can choose more than one answer)?			
Parents		812	88.3%
Spouse		697	75.8%
Siblings		505	54.9%
The patient has the right to reject the DNR decision:			
Agree		540	58.7%
Neutral		170	18.5%
Disagree		210	22.8%
What do you think are the factors to be considered when making the DNR decision?			
Religious concerns		811	88.2%
Legal concerns		774	84.1%
Risk of vegetative state		759	82.5%
Patient dignity		750	81.5%
Limited ICU state		529	57.5%
Efficient use of medical resources and cost reduction		491	53.4%
What is your degree of agreement with the following statements?			
	Agree	Neutral	Disagree
“It is acceptable to be conservative in investigations and treatments with patients who are labeled as DNR patients”.	45.5%	29.9%	24.7%
“It is acceptable to withdraw life sustaining treatment from DNR-labeled patients”.	23.5%	32.4%	44.1%
“The discussion of organ donation with DNR patients and/or their families should be encouraged”.	43.4%	39.8%	16.8%
“It is best that patients are not made aware of their DNR status”.	38.0%	27.0%	35.0%
“Discussion about the DNR order is stressful”.	59.6%	28.3%	12.2%

## Data Availability

The datasets used and/or analyzed during the current study are available from the corresponding author upon reasonable request.

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
