# Peer review of "The Perspective of the General Population in Saudi Arabia towards Do-Not-Resuscitate (DNR) Orders: A Cross-Sectional Study"

_healthcare, 2023, doi:10.3390/healthcare11142073_

Round 1
Reviewer 1 Report
General comment:
The authors investigated a cross-sectional study about the knowledge of DNR among the general population in Saudi Arabia and concluded that the knowledge of DNR is weak in their country. The perspective of this study is important to check and improve the awareness of DNR in their country. However, the reviewer should suggest severe concern about the content and structure of the manuscript.
Specific comments
Major comment 1:
The author should clearly describe what is already known, what is still unknown in this field, and what is the novel point of their study.
Major comment 2:
Since the questionnaire used in this study is not a widely used index worldwide, its accuracy is inevitably questionable for the readers. Can the authors provide more details about the results of the pilot study described in 2.5. to show some accuracy of the questionnaire?
Major comment 3:
The content of paragraph 2 in the Discussion part should be rewritten because it is mostly a repetition of the results and there is little new discussion.
Major comment 4:
Paragraphs 3-9 in the Discussion part are only reviews of past research, and their relevance to this research is unclear. The Discussion part of the research article should include discussions linked to the authors’ research results, so these parts need to be rewritten significantly.
Major comment 5:
The authors should describe the limitation of this study.
Major comment 6:
The reviewer thinks that their conclusion can be derived from the past studies they referred to. The authors describe the specific conclusion related to the result of the current study.
Minor comment 1:
The use of abbreviations is not unified. In general, the abbreviation should be defined the first time the word is used with full spelling out and then used consistently thereafter. In addition, the “DRN” may be the mistake of DNR (L18 and L51).
Minor comment 2:
In L44-L4, the sentence “because CPR's efficacy is greatly influenced by the patient's prognosis” is confusing, because the patient’s prognosis should be influenced by CPR’s efficacy controversially. Therefore the sentence should be rephrased by other words.
Minor comment 3:
In L48, “Alive” should be “alive”.
Minor comment 4:
Can the authors write a more detailed source for the questionnaire? E.x, the URL of the questionnaire, or the homepage where it is added (individual, hospital, or academic society?)
Minor comment 5:
Please specify the software used for statistical analysis in this study.
Minor comment 6:
All of the tables have inconsistent head positions of the sentence, which gives the difficulty to read.
Minor comment 7:
In Table 2, the question text is too long and difficult to read. It should be shorter.
Minor comment 8:
In Tables 1, 2, and 4, the usage of Frequency is incorrect. Frequency is not used for absolute numbers.
Minor comment 9:
Tables 1,2 should include the total number.
Minor comment 10:
In 3.1. and 3.2., the sentence “The Table. X presents…” should be placed top of the first paragraph for readability.
Minor comment 11:
In L144, the meaning of value 1.9 (1.3) is confusing. Is the “1.9” mean score? And then, what the “1.3” mean? Additionally, what is the evidence that “1.9” is a weak value? In other words, in this scoring system, what point is defined as strong, and what point is defined as weak?
Minor comment 12:
In 3.4, “Table 3” should be “Table 4”.
Minor comment 13:
Reference styles are not standardized. Please check the submission rules below.
Journal Articles:
1. Author 1, A.B.; Author 2, C.D. Title of the article. Abbreviated Journal Name Year, Volume, page range.
Author Response
General comment:
The authors investigated a cross-sectional study about the knowledge of DNR among the general population in Saudi Arabia and concluded that the knowledge of DNR is weak in their country. The perspective of this study is important to check and improve the awareness of DNR in their country. However, the reviewer should suggest severe concern about the content and structure of the manuscript.
Specific comments
Major comment 1:
The author should clearly describe what is already known, what is still unknown in this field, and what is the novel point of their study.
- Thank you for this comment, we have now addressed this comment and highlighted further that in Saudi Arabia, there are few studies that examined perception towards DNR, with the majority among patients with specific diseases such as cancer or explored perception of healthcare professionals and the importance of examining public perception towards DNR in the introduction in page 3.
Major comment 2:
Since the questionnaire used in this study is not a widely used index worldwide, its accuracy is inevitably questionable for the readers. Can the authors provide more details about the results of the pilot study described in 2.5. to show some accuracy of the questionnaire?
- Thank you for this comment, we have now addressed this comment and highlighted further the outcomes of the pilot study as the following: “The Arabic translated version of the questionnaire was checked by expert clinicians about its clarity and comprehensibility. They confirmed that the questionnaire items are clear and support the study objectives. This was followed by conducting a small pilot study on a group of participants from the general public. The questionnaire pilot study verified the following: the clarity and understandability of the survey items, their relevance and appropriateness to the research objectives, the appropriateness of the length of the questionnaire and the time required for completion, and the appropriateness of the response options and scale used.”, see page 4.
Major comment 3:
The content of paragraph 2 in the Discussion part should be rewritten because it is mostly a repetition of the results and there is little new discussion.
- Thank you for this comment, we have now addressed this comment and re-wrote and enriched our discussion section in pages 8-11.
Major comment 4:
Paragraphs 3-9 in the Discussion part are only reviews of past research, and their relevance to this research is unclear. The Discussion part of the research article should include discussions linked to the authors’ research results, so these parts need to be rewritten significantly.
- Thank you for this comment, we have now addressed this comment and re-wrote and enriched our discussion section in pages 8-11.
Major comment 5:
The authors should describe the limitation of this study.
- Thank you for this comment, we have now addressed this comment and added the study limitations in page 12.
Major comment 6:
The reviewer thinks that their conclusion can be derived from the past studies they referred to. The authors describe the specific conclusion related to the result of the current study.
- Thank you for this comment, we have now addressed this comment in page 12.
Minor comment 1:
The use of abbreviations is not unified. In general, the abbreviation should be defined the first time the word is used with full spelling out and then used consistently thereafter. In addition, the “DRN” may be the mistake of DNR (L18 and L51).
- Thank you for this comment, we have now addressed this comment throughout the manuscript.
Minor comment 2:
In L44-L4, the sentence “because CPR's efficacy is greatly influenced by the patient's prognosis” is confusing, because the patient’s prognosis should be influenced by CPR’s efficacy controversially. Therefore the sentence should be rephrased by other words.
- Thank you for this comment, we have now addressed this comment and rephrased the sentence in page 3.
Minor comment 3:
In L48, “Alive” should be “alive”.
- Thank you for this comment, we have now addressed this comment.
Minor comment 4:
Can the authors write a more detailed source for the questionnaire? E.x, the URL of the questionnaire, or the homepage where it is added (individual, hospital, or academic society?)
- Thank you for this comment, we have now addressed this comment in the method section under the subheading “Study design and settings”.
Minor comment 5:
Please specify the software used for statistical analysis in this study.
- Thank you for this comment, we have now addressed this comment in the statistical analysis section.
Minor comment 6:
All of the tables have inconsistent head positions of the sentence, which gives the difficulty to read.
- Thank you for this comment, we have now addressed this comment for all tables.
Minor comment 7:
In Table 2, the question text is too long and difficult to read. It should be shorter.
- Thank you for this comment, we preferred to keep the original definitions for DNR (available in table 2) provided in the original study from which we adapted pour questionnaire tool. We didn’t shorten the statements to keep them simple for the understanding of the general public who were the target population for our study.
Minor comment 8:
In Tables 1, 2, and 4, the usage of Frequency is incorrect. Frequency is not used for absolute numbers.
- Thank you for this comment. As the reviewer mentioned the use of frequency is not ideal for absolute number. Therefore, we used the percentage only while presenting our findings (in the text) and kept the number for the readers who are interested in viewing them (in the tables). The presentation of both frequency and percentages in the tables of the results section is a common practice for the vast majority of scientific research. However, if the reviewer prefers us to delete them, we can do so.
Minor comment 9:
Tables 1,2 should include the total number.
- Thank you for this comment, we have now addressed this comment and added the total number in the table title.
Minor comment 10:
In 3.1. and 3.2., the sentence “The Table. X presents…” should be placed top of the first paragraph for readability.
- Thank you for this comment, we have now addressed this comment.
Minor comment 11:
In L144, the meaning of value 1.9 (1.3) is confusing. Is the “1.9” mean score? And then, what the “1.3” mean? Additionally, what is the evidence that “1.9” is a weak value? In other words, in this scoring system, what point is defined as strong, and what point is defined as weak?
- Thank you for this comment, we have now clarified that 1.3 is the standard deviation for the mean score of the study sample. We defined any score below the midpoint of the maximum obtainable score which is 6 as poor knowledge. We have now clarified this point in the method section.
Minor comment 12:
In 3.4, “Table 3” should be “Table 4”.
- Thank you for this comment, we have now addressed this comment.
Minor comment 13:
Reference styles are not standardized. Please check the submission rules below.
Journal Articles:
1. Author 1, A.B.; Author 2, C.D. Title of the article. Abbreviated Journal Name Year, Volume, page range.
- Thank you for this comment, we have now checked our references list and we will make sure that references adhere to the guidelines of the journal in the production stage.
Reviewer 2 Report
The authors examined the knowledge and attitude toward do-not-resuscitate (DNR) among patients and their relatives visiting outpatient clinics in Saudi Arabi.
The article is important as it highlights a lack of knowledge about DNR in the general population.
Overall, the work is interesting, clear and well written.
However, I would suggest to the authors to expand some aspects to emphasize the results:
- Why are females, divorced and those who have post-graduate level more aware of the DNR? https://www.ncbi.nlm.nih.gov/pmc/articles/PMC9453860/
- Who are the participants with less knowledge of the DNR? Why?
The acronyms do-not-resuscitate (DNR) should be exposed in line 50 and not in line 56
Author Response
The authors examined the knowledge and attitude toward do-not-resuscitate (DNR) among patients and their relatives visiting outpatient clinics in Saudi Arabia.
The article is important as it highlights a lack of knowledge about DNR in the general population.
Overall, the work is interesting, clear and well written.
However, I would suggest to the authors to expand some aspects to emphasize the results:
- Why are females, divorced and those who have post-graduate level more aware of the DNR? https://www.ncbi.nlm.nih.gov/pmc/articles/PMC9453860/
- Thank you for this comment, we have now addressed this comment and discussed this point further using the above mentioned resource and other resources.
- Who are the participants with less knowledge of the DNR? Why?
- Thank you for this comment. We found that males were less likely to be knowledgeable about DNR. We have now discussed this point in the discussion section in page 9, as the following “In our study, females, divorced individuals, and those with a postgraduate degree were more likely than others to be familiar with DNR. In a previous study, Al Ahmadi et al. found that participants familiar with DNR had a high level of education, but there were no significant differences in gender, age, or marital status [15]. Another study by Vilpert et al. reported that females are better informed about end-of-life care options than males [19]. Besides, Vilpert et al. found that individuals with higher education levels have better knowledge about end-of-life care option [19]. Multiple factors could influence public knowledge of DNR such as gender, knowledge, and marital status. Higher levels of education (such as postgraduate degrees), are frequently accompanied by greater exposure to healthcare-related information. As a result, access to education and information as well as health literacy improve [19]. Other contributing factors may include personal experience and life circumstances and social factors, and the fact that females, divorced individuals, and those with higher levels of education could have greater opportunities to discuss DNR”.
The acronyms do-not-resuscitate (DNR) should be exposed in line 50 and not in line 56
- Thank you for this comment, we have now addressed this comment
Reviewer 3 Report
1. Information on how participants were recruited or selected should be provided to display the representativeness of the study sample. If a convenience sampling method was used, it may introduce bias and limit the generalizability of the findings to the broader population. Using a more rigorous sampling technique, such as random sampling, would enhance the external validity of the study.
2. The study does not mention any information regarding the validity and reliability of the data collection instruments used. It is crucial to establish the validity of the survey questionnaire or interview guide through pilot testing or validation studies. Additionally, reporting measures taken to ensure the reliability of the instruments, such as test-retest reliability or inter-rater reliability, is important to enhance the credibility of the findings.
3. Ethical considerations should be stated. It is essential to provide information on obtaining informed consent, ensuring participant confidentiality, and complying with ethical guidelines. The omission of these details raises concerns about the ethical integrity of the study and the protection of participants' rights.
4. The study focuses solely on a specific population in Saudi Arabia, without considering the diversity of cultural backgrounds or healthcare systems in other regions. This limits the generalizability of the findings to a broader context. Including a more diverse sample and conducting multi-center studies would enhance the external validity of the research.
5. Subgroup analysis is suggested, such as different age groups, educational levels, or geographic regions. Conducting subgroup analyses can provide valuable insights into variations or disparities in knowledge and attitudes towards DNR among different population segments.
6. The limitations section is inadequately addressed. It is essential to acknowledge the limitations of the study, such as potential biases, confounding factors, and any constraints that might have affected the study's outcomes. Furthermore, the discussion section should thoroughly discuss the implications of the findings, compare them with existing literature, and propose potential areas for future research. Expanding on these aspects will enhance the manuscript's overall quality.
The overall English writing in the manuscript needs improvement. There are several instances of grammatical errors, unclear sentence structures, and awkward phrasing. It is recommended to have a thorough proofreading and editing process to ensure the clarity and coherence of the manuscript.
Author Response
- Information on how participants were recruited or selected should be provided to display the representativeness of the study sample. If a convenience sampling method was used, it may introduce bias and limit the generalizability of the findings to the broader population. Using a more rigorous sampling technique, such as random sampling, would enhance the external validity of the study.
- Thank you for this comment. We have mentioned in the method section that we used convenience sampling technique. In order to highlight this point, we have now added it to the study limitations section in page 12 as the following “The use of convenience sampling technique and the online survey study design may have impacted the generalizability of our study's findings, as a substantial proportion of our target population may lack access to social media websites. Nevertheless, according to the most recent available data from 2023, approximately 79.3 percent of the Saudi Arabian population is active on social media.”.
- The study does not mention any information regarding the validity and reliability of the data collection instruments used. It is crucial to establish the validity of the survey questionnaire or interview guide through pilot testing or validation studies. Additionally, reporting measures taken to ensure the reliability of the instruments, such as test-retest reliability or inter-rater reliability, is important to enhance the credibility of the findings.
- Thank you for this comment. We have mentioned in the method section further details on the piloting study that was conducted in our research. However, unfortunately, we did not perform test-retest reliability or inter-rater reliability. We have now added this point to the study limitations section.
- Ethical considerations should be stated. It is essential to provide information on obtaining informed consent, ensuring participant confidentiality, and complying with ethical guidelines. The omission of these details raises concerns about the ethical integrity of the study and the protection of participants' rights.
- Thank you for this comment. All these details are available in page 12 as the following “Institutional Review Board Statement: This study was revised and approved by the Institutional Review Board at Al-Imam Muhammad Ibn Saud Islamic University, Riyadh, Saudi Arabia (Reference number: 426/2023). All methods were performed in accordance with the relevant guidelines and regulations. All participants gave their consent to participate in the study.”
- The study focuses solely on a specific population in Saudi Arabia, without considering the diversity of cultural backgrounds or healthcare systems in other regions. This limits the generalizability of the findings to a broader context. Including a more diverse sample and conducting multi-center studies would enhance the external validity of the research.
- Thank you for this comment. Our study population was the general public, which believe will increase the generalizability of our findings as previous study focused on specific populations such as healthcare professionals or patients with cancer or their relatives. We have now highlighted this point in the introduction section.
- Subgroup analysis is suggested, such as different age groups, educational levels, or geographic regions. Conducting subgroup analyses can provide valuable insights into variations or disparities in knowledge and attitudes towards DNR among different population segments.
- Thank you for this comment. The aim of this study was to examine the public knowledge of DNR. However, in order to address the reviewer comment, we conducted logistic regression analysis to identify demographic characteristics associated with higher likelihood of being knowledgeable about DNR and we identified that females, divorced individuals, and those with a postgraduate degree were more likely than others to be familiar with DNR and discussed this point in the discussion section.
- The limitations section is inadequately addressed. It is essential to acknowledge the limitations of the study, such as potential biases, confounding factors, and any constraints that might have affected the study's outcomes. Furthermore, the discussion section should thoroughly discuss the implications of the findings, compare them with existing literature, and propose potential areas for future research. Expanding on these aspects will enhance the manuscript's overall quality.
- Thank you for this comment. We have now added the study limitations and improved our discussing section based on the reviewer comment to enhance the quality of the manuscript.
Comments on the Quality of English Language
The overall English writing in the manuscript needs improvement. There are several instances of grammatical errors, unclear sentence structures, and awkward phrasing. It is recommended to have a thorough proofreading and editing process to ensure the clarity and coherence of the manuscript.
- Thank you for this comment. We have now addressed the reviewer comment and proofread our manuscript.
Round 2
Reviewer 1 Report
The authors revised the manuscript almost exactly according to the reviewer's comments. The revised manuscript is much clearer and better structured. The reviewers thank the authors for their revision efforts.
Minor comment 1
The Table.1 in 3.2 should be Table. 2.
Minor comment 2
the reference style is still slightly different from the Journal's style (ex. the volume is in italics, the issue is not necessary), and the authors should revise carefully the content again.
Author Response
Minor comment 1
The Table.1 in 3.2 should be Table. 2.
- In line 140, it is written as Table 2 not Table 1.
Minor comment 2
the reference style is still slightly different from the Journal's style (ex. the volume is in italics, the issue is not necessary), and the authors should revise carefully the content again.
- Thank you for this comment, we have now addressed this comment.
Reviewer 3 Report
The authors have addressed the comments satisfactorily.
Author Response
Thank you for confirming that you don't have any further comments.